# Assessment of Low-Level Air Pollution and Cardiovascular Incidence in Gdansk, Poland: Time-Series Cross-Sectional Analysis

**DOI:** 10.3390/jcm12062206

**Published:** 2023-03-13

**Authors:** Radosław Czernych, Artur Jerzy Badyda, Grzegorz Kozera, Paweł Zagożdżon

**Affiliations:** 1Department of Hygiene and Epidemiology, Faculty of Medicine, Medical University of Gdansk, 80-210 Gdansk, Poland; 2Department of Informatics and Environment Quality Research, Faculty of Building Services, Hydro- and Environmental Engineering, Warsaw University of Technology, 00-653 Warsaw, Poland; 3Centre of Medical Simulations, Medical University of Gdansk, 80-204 Gdansk, Poland

**Keywords:** air pollution, gaseous pollutants, particulate matter, cardiovascular disease, ischemic stroke, myocardial infarction

## Abstract

(1) Background: More than 1.8 million people in the European Union die every year as a result of CVD, accounting for 36% of all deaths with a large proportion being premature (before the age of 65). There are more than 300 different risk factors of CVD, known and air pollution is one of them. The aim of this study was to investigate whether daily cardiovascular mortality was associated with air pollutants and meteorological conditions in an urban environment with a low level of air pollution. (2) Methods: Data on daily incidence of strokes and myocardial infarctions in the city of Gdansk were obtained from the National Health Fund (NHF) and covered the period from 1 January 2014 to 31 December 2018. Data on the level of pollution, i.e., SO_2_, NO, NO_2_, NOx, CO, PM10, PM2.5, CO_2_, O_3_ and meteorological conditions came from the foundation: Agency of Regional Air Quality Monitoring in the Gdańsk metropolitan area (ARMAG). Using these data, we calculated mean values with standard deviation (SD) and derived the minimum and maximum values and interquartile range (IQR). Time series regression with Poisson distribution was used in statistical analysis. (4) Results: Stroke incidence is significantly affected by an increase in concentrations of NO, NO_2_ and NOx with RRs equal to 1.019 (95%CI: 1.001–1.036), 1.036 (95%CI: 1.008–1.064) and 1.017 (95%CI: 1.000–1.034) for every increase in IQR by 14.12, 14.62 and 22.62 μg/m^3^, respectively. Similarly, myocardial infarction incidence is significantly affected by an increase in concentrations of NO, NO_2_ and NOx with RRs equal to 1.030 (95%CI: 1.011–1.048), 1.053 (95%CI: 1.024–1.082) and 1.027 (95%CI: 1.010–1.045) for every increase in IQR by 14.12, 14.62 and 22.62 μg/m^3^, respectively. Both PM10 and PM2.5 were positively associated with myocardial infarction incidence. (5) Conclusions: In this time-series cross-sectional study, we found strong evidence that support the hypothesis that transient elevations in ambient PM2.5, PM10, NO_2_, SO_2_ and CO are associated with higher relative risk of ischemic stroke and myocardial infarction incidents.

## 1. Introduction

Myocardial infarction and ischemic stroke are mainly a complication of atherosclerotic lesions in the respective vessels (coronary or cerebral) and therefore have the same etiology and are both clinically defined as cardiovascular diseases (CVD) [1]. The burden of CVD is greater than that of any other disease and is the leading cause of death in Europe and in the world. More than 1.8 million people in the European Union die every year as a result of CVD, accounting for 36% of all deaths with a large proportion being premature (before the age of 65) [2]. According to WHO data, in 2003, premature cardiovascular mortality in Polish people aged 25–64 was 2.5 times higher than in other European Union countries. In 2014, cardiovascular diseases accounted for as much as 45.1% of all deaths in Poles, including 40.3% among men and 50.3% among women [3]. Myocardial infarction incidence differs depending on age and sex. In total, 121 men and 25 women per 100,000 between 40 and 44 years suffer from myocardial infarction. For older age groups, this value is multiplied and equals: 1012/416 (men/women aged 65–69) and 1718/1075 (men/women aged 80–84 years). Myocardial infarction hospital mortality in 2012 in Poland in patients aged 35–49 years was 2.5%; 60–64 years, 5%; 80–84 years, 15%; and over 85 years, exceeded 20% [4]. According to Global Burden of Disease data, the incidence rate in Poland for the first-in-life ischemic stroke in 2010 was 173.2/100,000, while two decades earlier, it was 186.6/100,000 [5]. According to the data of the National Health Fund, the number of all hospitalizations due to the treatment of stroke in Poland in 2009 was almost 95,000 and has decreased to over 88,000 since [6].

Although more than 300 different risk factors of these diseases are known, and incidence is gradually decreasing, both myocardial infarction and stroke remain a leading cause of disease disability and death in the Western world [7,8,9]. Aside from the most recognizable non-modifiable risk factors of CVD, there are factors defined as structural determinants, for example: living and working conditions. Both are often related with air pollution.

Air is a complex mixture of gases and aerosols. Its composition is usually affected by the degree of industrialization and urbanization in a given area [10]. Emissions of polluting agents such as sulfur dioxide (SO_2_), nitrogen dioxide (NO_2_), nitrogen oxide (NO), carbon monoxide (CO), particulate matter with a diameter of 10 μm or less (PM10) and of 2.5 μm or less (PM2.5) as well as benzo[a]pyrene can be mostly related with inefficient modes of transport (polluting fuels and vehicles), inefficient combustion of household fuels for cooking, lighting and heating, coal-fired power plants, agriculture, and waste burning [11]. The intensity of industrialization and urbanization affects not only the natural environment but also has a number of possible human health effects. The Great Smog of London took place in the early years of the second half of the 20th century and caused hundreds of thousands of hospitalizations followed by nearly 12,000 deaths [12]. After the incident, governments across the world began to understand that unsustainable development and lack of economic and ecological resource management can do more harm than good. Since that time, scientists, often supported by policy makers, began to investigate possible effects of a polluted environment on human health. Currently, air pollution is recognized as one of the risk factors, that on a massive scale, can cause: inflammation of the upper and lower respiratory track, lung cancers, cardiovascular diseases and preterm death [11].

Most of the studies investigating the effect of short-term exposure of air pollutants on the population concentrate on urban areas with high levels of air pollution, whereas there is a scarce amount of studies of the health effect of low-level pollution [13,14,15,16]. Because of its geographical location and characteristic climate condition, Pomorskie Voivodship, along with its capital city (Gdansk), is considered to have one of the cleanest atmospheric environments in the whole of Poland.

The aim of this study was to investigate whether daily cardiovascular mortality was associated with air pollutants such as: SO_2_, NO_2_, NO, NOx CO, O_3_, CO_2_ as well as PM10, PM2.5 and meteorological condition in an urban environment with low levels of air pollution. Separate analyses were carried out with respect to sex and age.

## 2. Materials and Methods

### 2.1. Study Area and Its Climate

Gdansk is a city localized on the southern coast of the Baltic Sea region (northern Poland). With a population of over 470,000 citizens and an area of 262 km^2^, it is the capital and largest city in Pomorskie Voivodship. In a conurbation with the city of Gdynia, the resort town of Sopot, and suburban communities, these form the metropolitan area called the Tri-City, with a population of approximately 1.5 million and an area of 414 km^2^. It is Poland’s principal seaport and the country’s fourth-largest metropolitan area [17].

Because of its characteristic geographic location, the Gdansk climate seems to have both oceanic and continental influences. According to different categorization systems, Gdansk either has an oceanic climate or belongs to the continental climate zone. The continental characteristic of Gdansk’s climate can be explained by dry winters and precipitation maximum during summer. Nonetheless, seasonal extremes are less pronounced than those in the inland parts of Poland. The average temperatures range from −1.0 to 17.2 °C, whereas average monthly precipitation varies from 17.9 up to 66.7 mm per month with an annual total of 507 mm [18].

The vicinity of the Baltic Sea, geomorphological diversity and location within the influence of large baric centers have a major impact on the speed and direction of winds in the Pomeranian Voivodeship. In the coastal zone of the voivodship, the dominant winds are west winds, while inland and in Żuławy, they are the southeast winds. In 2018, the majority of winds in the Pomeranian Voivodeship were southern winds, the average annual speed ranging from 3.1 to 5.1 m/s. Higher wind speeds occurred in the coastal strip of the Voivodship, while the most common silence was in the Tri-City area. In the Tri-City agglomeration, winds predominated, with the average annual speed ranging from 3.1 to 5.1 m/s. Higher wind speeds were most common on Sobieszewo Island in the coastal belt of Gdynia [18].

The main source of air pollution in the Pomeranian Voivodeship is anthropogenic emission. It is associated with point sources from industrial plants, mainly from fuel combustion processes for energy purposes and technological processes (23%), with a linear source related to road, rail, water and air transport (15%), as well as in the area, related to the municipal and housing sector (49%). In the case of point sources in the Tri-City agglomeration, the main pollutants are sulfur oxides SOx emitted by power plants and combined heat and power plants and, to a lesser extent, by production processes. On the other hand, in the Pomeranian zone, the highest share of sulfur oxides comes from households, i.e., from the municipal and living sector. The main pollutants emitted from road transport in the Pomeranian zone in 2018 were nitrogen oxides (NOx). In the case of the Tri-City agglomeration, the highest share in the emission of nitrogen oxides was point emission, mostly from heat and power plants in Gdańsk and Gdynia. Compact, low-rise buildings and the related heating processes in the municipal-living sector (surface emission) cause high concentrations of mainly suspended dust (PM10). Apart from households, the sources of such high emissions were also: agriculture, livestock farming, heaps, excavations, land and forests [19].

### 2.2. Incidence Data

Data on daily incidence of strokes and myocardial infarctions in the city of Gdansk were obtained from the National Health Fund (NHF) and covered the period from 1 January 2014 to 31 December 2018 (Table 1). The database contains information on the date of patient admission to the hospital unit and his/her discharge (or possible death), the poviat from which the patient came and the poviat of the hospital ward, sex and date of birth of the patient and the code of the diagnosed disease entity (ICD-10). In total, the database contains 73,413 cases of strokes and myocardial infarctions (ICD-10: I63, I61-62, respectively). Since the study focused on the area of the Gdansk municipality, cases of patients living in Gdańsk and admitted to the hospital unit in this area were selected from the NHF database. Finally, the numbers of strokes and infarctions each day were determined (total and broken down by gender and age groups), and the study period of daily incidence data was matched with the available period of daily environmental data for statistical analysis.

### 2.3. Environmental Data

Data on the level of pollution came from six monitoring stations located in Gdańsk and described the concentrations of the following chemical compounds: SO_2_, NO, NO_2_, NOx, CO, PM10, PM2.5, CO_2_, O_3_ and values of temperature, humidity, atmospheric pressure, rainfall and wind strength. The monitoring stations are owned and maintained by the foundation: Agency of Regional Air Quality Monitoring in the Gdańsk metropolitan area (ARMAG). Data collected were based on measurements carried out with 1 h intervals over the studied period (years: 2014–2018, for PM2.5 from 2015 to 2018). Apart from the missing measurement periods for PM2.5, there were also, unidentified as to the cause, missing values. The number of missing values is presented in Table 2. In order to obtain a single concentration level for an individual compound for one day for the entire city of Gdańsk, the measurements from all stations were averaged, and then the third quartile (75%) was selected as a measurement representative for each day. All further analyses are based on daily observations conducted for the studied 5-year period (total count of observations: 1825).

### 2.4. Statistical Methods

Descriptive statistical analyses of mortality data and environmental data are summarized into mean, standard deviation, and maximum, minimum, and interquartile range (IQR) in Table 1. In the case of strokes and infarctions, instead of the number of missing observations, we give the total number of occurrences and the median.

**Table 1 jcm-12-02206-t001:** Summary of daily hospital admission for strokes (I63) and myocardial infarctions (I21, I22) in the Gdansk agglomeration in the years 2014–2018.

	Total Count (% of All)	Daily			
		Mean (SD)	Min.	Max.	IQR
Strokes (ICD-10: I63)	7619	3.48 (1.99)	0	13	3
Women	4047 (53%)	1.85 (1.45)	0	9	2
Men	3572 (47%)	1.63 (1.35)	0	8	1
After 65 years	5520 (72%)	2.52 (1.68)	0	11	3
Before 65 years	2099 (28%)	0.96 (1.03)	0	6	2
Myocardial-infarctions (ICD-10: I21, I22)	6910	3.15 (1.96)	0	12	2
Women	2750 (40%)	1.26 (1.18)	0	7	2
Men	4160 (60%)	1.90 (1.50)	0	9	2
After 65 years	4061 (59%)	1.85 (1.44)	0	9	2
Before 65 years	2849 (41%)	1.30 (1.27)	0	8	2

SD: standard deviation, Min: minimum, Max: maximum, IQR: interquartile range.

The Pearson correlation matrix was used to assess the relationships between the exposure factors.

The essential element of the analysis presented below is the assessment of the influence of pollution and meteorological conditions on the daily number of strokes and infarctions. An appropriate statistical model to study such a phenomenon is Poisson regression.

The Poisson regression model is described by the equation:log (E (Y)) = β0 + β1X1 + … + βnXn,
where Y is the dependent variable, depending on the explanatory variables X1, …, Xn, E (Y) is the average daily number of strokes or infarctions (in the Poisson distribution interpreted as intensity), while X1, …, Xn are the exposure factors (chemical compounds and weather conditions).

One of the basic assumptions of this model is that the resulting variable follows the Poisson distribution. Therefore, before starting the regression analysis, Chi-square tests were performed for total strokes and infarctions as well as in the groups of women and men, and people over and before the age of 65; additionally, the frequency charts were compared with the theoretical Poisson distributions. In all cases, the distributions were not significantly different from the Poisson distribution (in each case, the *p* value was close to 1).

First, we built regression models for individual exposure factors, taking into account the so-called lag (0–3 days) for the entire population, and then simple regression models (with one variable) in the subgroups of admitted patients—women/men and people over/before 65 years of age. Finally, we presented a multiple regression model for 9 chemical compounds (SO_2_, NO, NO_2_, NOx, CO, PM10, PM2.5, CO_2_, O_3_) as well as for temperature, humidity and atmospheric pressure.

On the basis of the above models, we obtained relative risk assessments related to the increase in the concentration level of pollutants with IQR and 10 degrees Celsius for temperature, 5% for humidity and 5 hPa for atmospheric pressure. The air pollutants with relative risks (RRs) and 95% confidence intervals for IQR change in pollution levels less than a significance level of 0.05 in the single-pollutant model were considered for our models.

All calculations and graphs were made using the R statistical package (version 3.4.1).

**Table 2 jcm-12-02206-t002:** Summary of daily environmental conditions in Gdansk agglomeration between 1 January 2014 and 31 December 2018.

	Missing Values (Total Count for Studied Period)	Daily			
		Mean (SD)	Min	Max	IQR
Chemical compounds (μg/m^3^)					
SO_2_	0	6.31 (4.08)	1.98	57.73	3.65
NO	0	22.83 (17.57)	4.46	170.94	14.12
NO_2_	0	23.55 (11.96)	5.09	96.49	14.62
NOx	0	36.47 (29.65)	7.05	294.11	22.52
CO	0	496.1 (203.26)	244.3	2280.1	164.04
PM10	0	26.87 (16.68)	5.66	151.17	16.76
PM2.5	393	20.07 (14.27)	3.58	178.83	12.84
O_3_	2	55.53 (22.8)	2.36	130	31.7
CO_2_	8	800.6 (60.64)	681.9	1094	79.89
Meteorological data					
Temperature (°C)	0	10.32 (8.73)	−15.99	33.08	13.7
Atmospheric pressure (hPa)	0	1011.4 (8.51)	978.2	1039.5	10.95
Humidity (%)	0	83.05 (7.14)	46.18	96.84	9.46

SD: standard deviation, Min: minimum, Max: maximum, IQR: interquartile range.

## 3. Results

Table 1 and Table 2 summarize the daily incidence and data on environmental conditions used in the models. During the period from 1 January 2014 to 31 December 2018, there were 7619 cases of strokes. On average, there were more than three incidents of stroke each day (with maximum value of 13 strokes). The majority of cases were women (53%) and men (47%) older than 65 years (72%). During the same period, there were 6910 myocardial infarctions. The majority of cases were men (60%) and people above 65 years of age (59%).

During the study period, daily temperatures fluctuated between −16 and 33 °C, and ambient air pollutant levels ranged from 5.1 to 96 μg/m^3^ for NO_2_, from 244 to 2280 μg/m^3^ for CO, and from 5.7 to 151 μg/m^3^ for PM10. The IQR of the environmental data, which were used for calculating RRs in the GAM, was 13.7 °C for daily temperature, 14.6 μg/m^3^ for NO_2_, 164.04 for CO, and 16.76 μg/m^3^ for PM10.

Table 3 shows the Pearson correlation coefficients between gaseous compounds such as SO_2_, NO, NO_2_, CO, particulate matter (PM2.5, PM10) and meteorological conditions such as temperature, humidity and pressure. The gaseous pollutants along with particulate matter were highly correlated with each other. The correlation ranged from 0.62 to 0.96. Correlation between PMs and CO was strongest (r_PM10_ = 0.79, r_PM2.5_ = 0.85). Statistically comparable was the correlation of PMs and NOx (r = 0.73–0.76). SO_2_ correlated with PMs the least (r_PM10_ = 0.62, r_PM2.5_ = 0.64). However, meteorological conditions such as temperature were moderately correlated with concentrations of traffic-related pollutants. Moreover, this correlation most often seems to be inverse, i.e., SO_2_ and CO were strongest (r = −0.49), whereas NO_X_ and PMs correlations ranged from −0.29 to −0.21.

The associations between stroke and myocardial infarction incidence and air pollutants with respect to lagged time are presented in Figure 1 and Figure 2, respectively. By adding individual air pollutants to the single-pollutant model, it could be observed that NO, NO_2_ and NOx with lag 0 were significantly associated with stroke incidence. RRs for NO, NO_2_ and NOx are 1.025, 1.044 and 1.022, respectively.

The same pollutants are significantly related with an increased incidence of myocardial infarction. RRs for NO, NO_2_ and NOx are 1.031, 1.052 and 1.029, respectively. Furthermore, SO_2_ and PM2.5 both lagged 2 days, which were significantly related with an increased incidence of myocardial infarction as well.

Although the RRs for stoke and myocardial infarction were not significant for other pollutants and lags, RRs for NO, NO_2_, NOx, SO_2_ as well as for PM10 and PM2.5 were mostly greater than 1 for lags 0 and 1.

Relative risks for the whole studied population with subdivision with respect to sex and age are presented in Table 4. Stroke incidence is significantly affected by an increase in concentrations of NO, NO_2_ and NOx with RRs equal to 1.02 (95%CI: 1.001–1.04), 1,04 (95%CI: 1.01–1.06) and 1,02 (95%CI: 1.00–1.03), respectively. Visibly, the relationship between NO_2_ and stroke incidence is stronger than for other nitrogen oxides. Regarding the selected subgroups, the same relationship can be observed in the cases of women and people aged 65 year and more. For women, RRs for NO and NO_2_ are higher than for the general population and are equal to 1.02 (95%CI: 1.00–1.05) and 1.05 (95%CI: 1.01–1.09). Further analysis of the selected age groups shows distinct vulnerability to nitrogen oxides in the group of elderly people (≥65 y.o.). RRs for NO and NO_2_ were the same with women. The elderly group, however, shows much a higher and significant susceptibility to NOx as well as to PM2.5. The effect of the latter was not statistically significant in any other subpopulation. Although it was not statistically significant, the RRs for NO, NO_2_, NOx as well as for PM10 and PM2.5 were, in general, higher than 1 in the whole population as well for all four subgroups. Myocardial infarction incidence is positively associated with more atmospheric air pollutants than ischemic strokes. The increase in relative risk was 1.03 (95%CI: 1.01–1.05) per 10 μg/m^3^ increase in SO_2_ concentration. Weaker but significant was the association of carbon monoxide, 1.02 (95%CI: 1.00–1.04).

Similarly, myocardial infarction incidence is significantly affected by an increase in concentrations of NO, NO_2_ and NOx with RRs equal to 1.03 (95%CI: 1.01–1.05), 1.05 (95%CI: 1.02–1.08) and 1.03 (95%CI: 1.01–1.05), respectively. Both PM10 and PM2.5 were positively associated with myocardial infarction incidence, with a stronger association for PM2.5. The increase in RR was 1.03 (95%CI: 1.01–1.06) per 10 μg/m^3^ increase in PM2.5 concentration, whereas RR for PM10 was 1.03 (95%CI: 1.00–1.05). Analysis of sex and age shows that the association between myocardial infarction and nitrogen oxide increase in concentration persists in all subgroups but seems to be stronger for women (1.06 (95%CI: 1.01–1.11) per 10 μg/m^3^ increase in NO_2_) and people 65 years old or older (1.07 (95%CI: 1.02–1.16) per 10 μg/m^3^ increase in NO_2_). Nonetheless, statistical significance of the association of infarction and NO and NOx persisted only for men. Men seem to be more vulnerable to particulate matter as well. PM10 and PM2.5 were positively associated with myocardial infarction incidence, with a stronger association for PM10. The increase in RR was 1.04 (95%CI: 1.01–1.07) per 10 μg/m^3^ increase in PM10 concentration, whereas RR for PM2.5 was 1.04 (95%CI: 1.01–1.07). Strength of association between myocardial incidence and particulate matter for the elderly group was significant for both PM10 and PM2.5 but with a stronger effect of PM2.5 (1.04 (95%CI: 1.00–1.08) per 10 μg/m^3^).

Aside from atmospheric pollutants, meteorological factors seem to play a certain role in the incidence of stroke. A positive association between temperature and strokes is significant for the young group of the studied population (<65 y.o.) and was equal to: 1.06 (95%CI: 1.01–1.11) for every 10 °C increase. Although not significant, this relationship can also be defined as strong for the group of men (1.02 (95%CI: 0.98–1.06) for every 10 °C increase. Humidity, on the other hand, is positively and significantly related with myocardial infarction, with the RR equal to 1.03 (95%CI: 1.01–1.04) for every 5% increase in humidity. The association persisted in both sex and age subgroups.

## 4. Discussion

According to our findings, the association between particulate pollutants (PM2.5 and PM10) and stroke was positive but not statistically significant. In the case of myocardial infarction, the positive association reached a level of significance. Those results were consistent with a number of studies that reported a positive association between PM2.5 or PM10 and relative risk of CVD in general or specific cardiovascular disease, i.e., stroke (ischemic/hemorrhagic), ischemic heart disease, myocardial infarction or atrial fibrillation [13,20,21,22,23]. In the study of two cohorts (the PPS cohort and the GOT-MONICA cohort), Stockfelt et al. received a positive association between both PM10 and PM2.5 for ischemic heart disease and heart failure for only one of the cohorts, whereas the results for the other cohort were insignificant [23]. Our results are in accordance with the findings of Kuźma et al., who observed a significant effect of particulate matter exposure to cardiovascular disease hospitalizations as well as to CVD deaths [24,25].

We observed a positive association between NO_2_, NO and NO_x_ and stroke as well as to myocardial infarction incidence. These findings were in agreement with a meta-analysis of 94 studies of 6.2 million events, across 28 countries, which found that NO_2_, SO_2_, and CO were all linked with higher risk of total stroke hospitalizations [26]. Our findings are also in line with Chan et al., who in a study based on hospital admission of National Taiwan University Hospital (NTUH), showed that NO_2_ was positively associated with higher risk of stroke mortality in China [27]. Our findings are also consistent with a large body of prior studies that assessed the positive relationship between stroke subtypes and air pollution [20,22,28]. However, other studies failed to find a positive association between gaseous pollutants and ischemic stroke or myocardial infarction [16,29,30,31].

The effect of SO_2_ on the incidence of CVD was pronounced and statistically significant only in the case of myocardial infarction both in female and male populations and elderly people. The effect was more pronounced in the group of women. These results are in agreement with Khniebadi et al., who assessed an increased risk of 2.7% of acute myocardial infarction among 540.000 citizens of Khorramabad, Iran [32]. The effect of SO_2_ in the risk of myocardial infarction was stronger in the study conducted by Tuan et al. in the Brazilian municipality of São José dos Campos [33]. Filho et al. observed an increase in interquartile range within a 2-day moving average of 8.0 µg/m^3^ of SO_2_ that was associated with 7.0% and 20.0% increases in cardiovascular disease emergency room visits by non-diabetic and diabetic groups, respectively [34]. Similar results were obtained in a study conducted in China (Hefei) [35].

In our assessment, a statistically significant effect of carbon monoxide was limited to myocardial infarction incidence, whereas stroke incidence of women and elderly people was substantially increased but still below the significance level. Most of the prior studies are in line with our assessment. Liu et al., in a time-series analysis in 272 cities in China, found significant associations between short-term exposure to ambient carbon monoxide and cardiovascular disease mortality in China [36]. An increase in percentage mortality due to CVD was also assessed in another time-series study in China [37]. In a time-stratified case-crossover analysis conducted by Son et al., the effect of CO toward CVD mortality was highest from all air pollutants assessed by the authors, i.e., PM10, NO_2_, SO_2_ and CO [38]. The same observations were also made by other authors [20,39].

According to our assessment, ozone has a negative significant association with the incidence of myocardial infarction. There was no significant relationship between ozone and stroke incidence. To the best of our knowledge, there are two possible explanations. First, there were only two monitoring stations that assessed ozone concentrations that may not cover all fluctuations of ozone atmospheric concentrations. These results can also be explained by non-linear association of ozone and CVD morbidity or mortality with a threshold from 25 to 60 μg/m^3^ [36,40,41,42]. Numerous studies seem to be in line with our findings [43,44,45]. According to the results from a Danish cohort study of 49,564 individuals, the relationship of ozone and CVD mortality was inverse. The research team pointed out that O3 is often inversely correlated with pollutants such as NO_2_ and PM due to the atmospheric chemical reaction between O_3_ and NO forming NO_2_; the inverse correlation is also evident in our data [43]. The same statement was made by other authors who received the same curing effect of ozone [15,46]. Other long-term studies suggest minor or no significant effect on CDV risk of morbidity or mortality [41,47,48]. On the other hand, there is a large body of evidence that ozone increases the risk of CVD mortality [36,40,41,42,49,50,51,52].

According to our lag analysis, NO, NO_2_ and NOx lag 0 were significantly associated with stroke incidence, while lags 1–3 were not related with increased risk for stroke or for myocardial infarction. Moreover, SO_2_ lagged 2 days and PM2.5 lagged 2 days were significantly related with an increased incidence of myocardial infarction. Although the RRs for stoke and myocardial infarction were not significant for other pollutants and lags, RRs for NO, NO_2_, NOx, SO_2_ as well as for PM10 and PM2.5 were mostly greater than 1 for lags 0 and 1. The results suggest that the effects of PM2.5 are acute in terms of CVD deaths. Other studies discovered similar findings. Results from a national study in the United States showed that on lag day 0 and 1, PM2.5 had the largest effect on CVDs [53]. Another study in Beijing also discovered PM2.5′s lag effects within 0–3 lag days [54]. According to the lag analysis of Tian et al., the lag association of PM2.5 with CVD mortality was statistically significant from lag day 0 to 3 [55]. In our study, the strongest association between PM10 atmospheric concentration and incidence of stroke and myocardial infarction occurred for lags 0 and 1. Similar results were assessed by Fisher et al. According to the results of his case-crossover analysis, the strongest association of PM10 on ischemic stroke was for the lag 0 period [13]. This association was also observed by other authors [51,56,57,58].

Our age-oriented analysis of the influence of air pollutants indicated that elderly populations are among the most vulnerable to the risk of stroke and myocardial infarction. Risk of stroke and myocardial infarction for this group was visibly increased for the exposure to NO_2_, NO and NOx and PM2.5, whereas exposure to SO_2_ and PM10 was related with an increased incidence of myocardial infarction. Statistical significance was not achieved for any of air pollutants studied for stroke incidence for the younger part of the studied population. On the other hand, exposure of the younger group to NO_2_, NO and NOx has a substantially increased incidence of myocardial infarction. Similar findings were observed by Goldberg et al., who found positive associations between daily non-accidental mortality of the elderly group population and all air pollutants except for O_3_ [59]. This specific susceptibility of elderly people was also emphasized by other researchers [26,60,61,62,63,64,65].

Evidence suggests that women may be more vulnerable than men to develop cardiovascular events upon air pollution exposure [23,35,59,66,67,68,69,70,71]. In our research, stroke incidence due to exposure to NO_2_ and NO was statistically significant only in the women group. An influence of PM2.5 on stroke incidence among women, although insignificant, should be noted. Due to Clougherty’s review, more studies of adults report stronger effects among women, particularly for older persons [72]. In the PAPA Study (The Public Health and Air Pollution in Asia), Kan et al. reported stronger associations between the gaseous pollutants SO_2_, NO_2_, O_3_ and PM10 and daily respiratory mortality among women and the elderly [73]. On the other hand, there are some that have reported no gender differences or that have found stronger effects in men [65,74,75,76,77].

Results obtained for carbon dioxide are confusing. Evidence presented in the novel literature shows that prolonged exposure to higher than normal concentrations of CO_2_ in atmospheric air can cause a variety of negative health effects. These may include headache, dizziness, restlessness, tingling or pins or needles feeling, difficulty breathing, sweating, tiredness, increased heart rate, elevated blood pressure, coma, asphyxia, and convulsions [78,79]. Nonetheless, the relationship between low-level exposure to CO_2_ and the reported symptoms remains inconsistent between studies [80,81]. Furthermore, the literature is missing novel studies concerning short-term CO_2_ exposure with respect to the risk of either stroke or myocardial infarction. According to the literature, carbon dioxide (CO_2_) levels affect vascular smooth muscle tone and hence play an important role in cerebral autoregulation [82,83,84]. Based on the evidence provided by Slinet et al.’s meta-analysis, acute stroke patients are significantly more likely, compared with controls, to be hypocapnic [85]. Hypocapnic patients suffer from reduced carbon dioxide concentration in the blood. Physiological concentration of carbon dioxide plays a role in relaxing the bronchi, smooth muscles, airways and blood vessels, increasing their diameter and thus reducing the risk of either stroke or myocardial infarction [86]. These physiological findings seem to be in accordance with our assessment results. Nonetheless, further epidemiological studies are required in order to evaluate the clinical impact of these findings.

Our study has some important limitations related to exposure assessment. First, we used averaged estimates of ambient air pollution from six stations situated at different locations of the Gdansk municipality area, as precise addresses of hospitalized patients were not accessible. Therefore, actual personal exposure might differ from the values calculated and used for this assessment. Second, the ordinary kriging method we used to estimate daily air pollutants likely reflects urban-scale variation in pollutant levels but may not fully capture microscale spatial gradients typical of urban environments. Moreover, air pollution in urban areas is characterized by high spatial fluctuations in the levels of pollutants, which could affect the results. Third, as we used incidence data coming from the National Health Fund, such a database may contain some disease misclassifications and gaps which we do not know about. Moreover, differences between the day of stroke symptom onset and day of hospitalization may have introduced some degrees of exposure misclassification, which may tend to bias the risk estimates toward the null. Fourth, in regression models, the observations must be independent, which in our case would mean that strokes and myocardial infarctions occurring on a given day do not affect the likelihood of these occurring in the future (within the studied period). This assumption is clearly wrong, as such an incident clearly increases the likelihood of it recurring in the future, and a repeat stroke might not be related to exposure factors but simply to patient compliance. Fortunately, the share of reoccurring myocardial infarctions in our database was only 0.3% (22 out of 6910 cases). Therefore, we treat these infarctions as ordinary cases, and regression models are created for all types of infarction.

## 5. Conclusions

In this time-series cross-sectional study, we found strong evidence that supports the hypothesis that transient elevations in ambient PM2.5, PM10, NO_2_, SO_2_ and CO are associated with higher relative risk of ischemic stroke and myocardial infarction incidents. Daily levels of NO_2_ and NOx were significantly associated with the relative risk of ischemic stroke, although we did not find such significance for PM10 and PM2.5. On the other hand, strong associations of particulate matter as well as gaseous air pollutants mentioned above were observed for myocardial infarction. The strongest health outcome of pollutants exposure was observed within the same day and diminished with every following day. Health effects were heterogeneous with respect to age and sex. To the best of our knowledge, this is the first short-time assessment of CVD incidence due to the exposure to air pollutants for a seaside region with a low level of air pollution, and as such, the results can be extrapolated to similar locations. Nonetheless, the health effects of the above pollutants with respect to other cerebrovascular, cardiovascular and cardiopulmonary diseases are scarce and require further research.

## Figures and Tables

**Figure 1 jcm-12-02206-f001:**
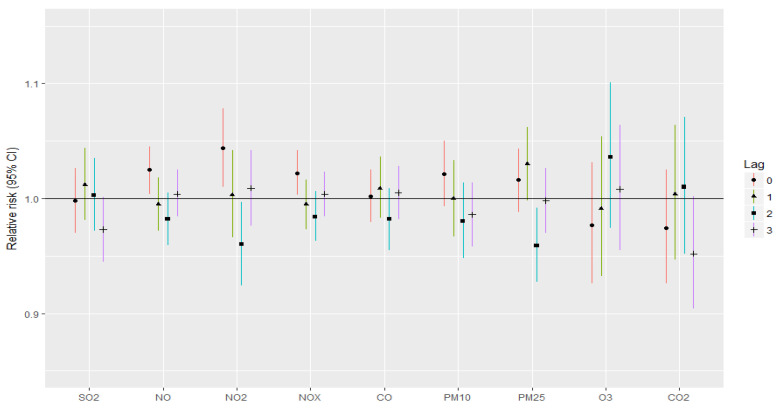
Relative risk of stroke for IQR changes in air pollutant levels for different lags in single-pollutant models.

**Figure 2 jcm-12-02206-f002:**
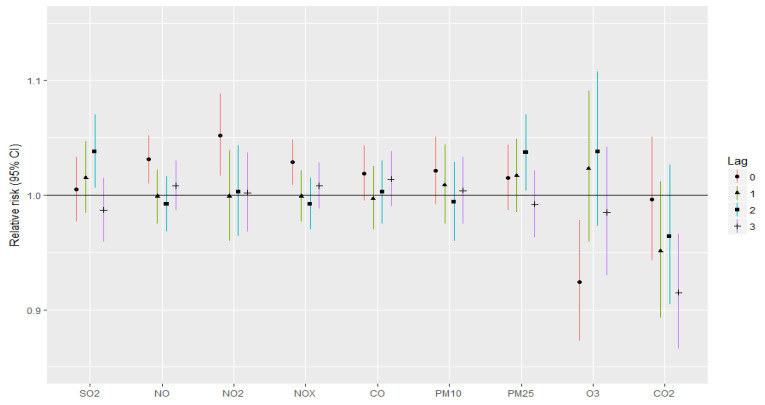
Relative risk of myocardial infarction for IQR changes in air pollutant levels for different lags in single-pollutant models.

**Table 3 jcm-12-02206-t003:** Pearson correlation coefficients for air pollutants and meteorological variables. (* *p* value < 0.05).

	SO_2_	NO	NO_2_	NOx	CO	PM10	PM2.5	O_3_	CO_2_	Temp.	Humid	Pres
SO_2_	1											
NO	0.47 *	1										
NO_2_	0.56 *	0.88 *	1									
NOx	0.46 *	1 *	0.86 *	1								
CO	0.61 *	0.86 *	0.79 *	0.85 *	1							
PM10	0.62 *	0.73 *	0.75 *	0.76 *	0.79 *	1						
PM2.5	0.64 *	0.75 *	0.75 *	0.74 *	0.85 *	0.96 *	1					
O_3_	−0.21 *	−0.37 *	−0.26 *	−0.38 *	−0.45 *	−0.20 *	−0.33 *	1				
CO_2_	0.03	0.32 *	0.29 *	0.32 *	0.34 *	0.30 *	0.23 *	−0.09 *	1			
Temp.	−0.49 *	−0.29 *	−0.29 *	−0.29 *	−0.49 *	−0.21 *	−0.33 *	0.54 *	0.01	1		
Humid	−0.01	0.21 *	0.13 *	0.21 *	0.22 *	0.02	0.16 *	−0.48 *	−0.02	−0.13 *	1	
Pres	0.13 *	0.15 *	0.16 *	0.15 *	0.15 *	0.20 *	0.18 *	0.04 *	0.15 *	−0.08 *	−0.19 *	1

**Table 4 jcm-12-02206-t004:** Relative risk of stroke and myocardial infarction for IQR changes in air pollutant level, 10 °C in temperature, 5% in humidity and 5 hPa in atmospheric pressure obtained from simple regression models (with one factor).

	SEX		AGE		ALL
	WOMEN	MEN	≥65 YEARS	<65 YEARS	
**STROKE**					
SO_2_	1.00 (0.97–1.03)	0.99 (0.96–1.02)	1.01 (0.99–1.03)	0.95 (0.91–0.99) *	0.99 (0.97–1.01)
NO	1.02 (1.00–1.05) *	1.01 (0.99–1.04)	1.02 (1.00–1.04) *	1.06 (0.97–1.04)	1.02 (1.00–1.04) *
NO_2_	1.05 (1.01–1.09) *	1.02 (0.98–1.06)	1.05 (1.06–1.08) *	1.00 (0.95–1.06)	1.04 (1.01–1.06) *
NOx	1.02 (1.00–1.04)	1.01 (0.99–1.04)	1.02 (1.00–1.04) *	1.01 (0.97–1.04)	1.02 (1.00–1.03) *
CO	1.01 (0.99–1.04)	0.99 (0.97–1.02)	1.01 (0.99–1.03)	0.98 (0.94–1.01)	1.00 (0.98–1.02)
PM10	1.02 (0.99–1.05)	1.00 (0.97–1.04)	1.02 (0.99–1.05)	0.99 (0.94–1.03)	1.01 (0.99–1.03)
PM2.5	1.02 (0.99–1.05)	1.01 (0.98–1.04)	1.03 (1.00–1.05) *	0.99 (0.94–1.03)	1.02 (0.99–1.04)
O_3_	0.98 (0.94–1.03)	1.02 (0.97–1.07)	0.98 (0.96–1.04)	1.01 (0.95–1.07)	1.00 (0.97–1.03)
CO_2_	0.95 (0.91–0.99) *	0.96 (0.92–0.99) *	0.96 (0.92–0.99) *	0.94 (0.89–0.99) *	0.95 (0.92–0.98) *
Temp.	0.98 (0.95–1.02)	1.02 (0.98–1.06)	0.98 (0.95–1.01)	1.06 (1.01–1.11) *	0.99 (0.97–1.02)
Humid.	1.00 (0.98–1.02)	0.98 (0.96–1.01)	0.98 (0.96–1.00)	1.02 (0.99–1.05)	0.99 (0.98–1.01)
Pres.	1.00 (0.99–1.02)	0.99 (0.98–1.01)	1.01 (0.99–1.02)	0.98 (0.95–1.00)	1.00 (0.99–1.01)
**MYOCARDIAL INFARCTION**					
SO_2_	1.04 (1.00–1.07) *	1.03 (1.00–1.05) *	1.04 (1.01–1.07) *	1.02 (0.99–1.05)	1.03 (1.01–1.05) *
NO	1.03 (0.99–1.06)	1.03 (1.01–1.06) *	1.04 (1.01–1.07) *	1.02 (1.00–1.05) *	1.03 (1.01–1.05) *
NO_2_	1.06 (1.01–1.11) *	1.05 (1.01–1.09) *	1.07 (1.02–1.16) *	1.04 (1.00–1.08) *	1.05 (1.02–1.08) *
NOx	1.02 (0.99–1.05)	1.03 (1.01–1.05) *	1.03 (1.01–1.06) *	1.02 (1.00–1.04) *	1.03 (1.01–1.04) *
CO	1.00 (0.97–1.03)	1.04 (1.01–1.06) *	1.03 (0.99–1.06)	1.02 (0.99–1.04)	1.02 (1.00–1.04) *
PM10	1.00 (0.96–1.04)	1.04 (1.01–1.07) *	1.04 (1.00–1.07) *	1.02 (0.99–1.05)	1.02 (1.00–1.05) *
PM2.5	1.03 (0.99–1.07)	1.04 (1.01–1.07) *	1.04 (1.00–1.08) *	1.03 (0.99–1.06)	1.03 (1.01–1.06) *
O_3_	0.94 (0.89–0.99) *	0.97 (0.93–1.01)	0.96 (0.91–1.01)	0.95 (0.91–0.99) *	0.96 (0.92–0.99) *
CO_2_	0.85 (0.81–0.90) *	0.90 (0.86–0.93) *	0.84 (0.80–0.88) *	0.91 (0.87–0.94) *	0.88 (0.85–0.91) *
Temp.	0.94 (0.90–0.98) *	0.96 (0.93–0.99) *	0.97 (0.93–1.01)	0.95 (0.91–0.98) *	0.95 (0.93–0.98) *
Humid.	1.02 (0.99–1.05)	1.03 (1.01–1.05) *	1.04 (1.01–1.06) *	1.02 (0.99–1.04)	1.07 (1.01–1.04) *
Pres.	0.98 (0.96–1.01)	0.99 (0.97–1.01)	0.99 (0.97–1.01)	0.98 (0.98–1.00)	0.98 (0.97–0.99) *

* *p* value < 0.05.

## Data Availability

The data presented in this study are available upon request from the corresponding author.

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
