# Peer review of "Assessment of Low-Level Air Pollution and Cardiovascular Incidence in Gdansk, Poland: Time-Series Cross-Sectional Analysis"

_jcm, 2023, doi:10.3390/jcm12062206_

Round 1

Reviewer 1 Report

Review for the MS “Assessment of Low-Level Air Pollution and Cardiovascular Incidence in Gdansk, Poland: time-series cross-sectional analysis”

The MS is dedicated to investigating the relationship between low-level air pollutants and the major cardiovascular diseases, myocardial infarction, and stroke in the Gdansk area.  

Abstract: Please provide us with what air pollutant measurement you use in this investigation. Average values, maximal values, relative numbers (x normal values), cumulative burden, or what.

Are these associations with stroke and myocardial infarction adjusted to age, gender, arterial hypertension, hypercholesterolemia, smoking habit, diabetes…

In introduction line 48 – hospital mortality for what? MI?

Units used in the climate description need review from the meteorologist.

What is a source of these data?

 The main source of air pollution in the Pomeranian Voivodeship is anthropogenic 119

emission. It is associated with point sources from industrial plants, mainly from fuel com- 120

bustion processes for energy purposes and technological processes (23%), with a linear 121

source related to road, rail, water, and air transport (15%), as well as in the area, related to 122

the municipal and housing sector (49%). In the case of point sources in the Tri-City ag- 123

glomeration, the main pollutants are sulfur oxides SOx emitted by power plants and com…- 124

137 – obtained (error in writing).

I don’t understand this:

Measurements were carried 1 hour over the years 2014- 154

2018, except for: PM2.5 (from 2015). Apart from the missing measurement periods, there 155

are also a few random, unidentified as to the cause, missing values, the number of which 156

is included and given later in the report. In order to obtain a single concentration level of 157

individual compounds for the entire city of Gdańsk, the measurements from station 158

were averaged, and then the third quartile (75%) was selected from among 24 hourly re- 159

results as a measurement representative for each day.

This is special statistic and needs advanced statistician to review it!

I don’t understand table 1, what is mean, minimum, median, maximum, IQR?

I also don't understand the values of pollutants in table 2. Please describe above the table what the figures mean. Average concentrations during the year? How many measurements during the year are done? Were there any large differences during winter, summer, wind, and rain???

I don’t understand figures 3 and 4. Please explain in a simple way what is presented.

I think that this MS needs more explanations to be clear and simpler.

Please do so, and after that, I can better review the article. 

Reviewer 2 Report

This is a cross-sectional analysis for a small town in northern Poland. 

To my mind the introduction of the article could be shortened by removing the obvious information about the causes of MI and stroke.

I think that Figure 2 does not carry any important semantic load.

It is  unclear to me why the authors didn’t comment the received data of the CO2 parameter. After all, in the table 4 there is a clear connection with both stroke and myocardial infarction, this connection is both for men and women, and for all ages. And according to these figures  CO2 plays a protective role. Is that so?

Round 2

Reviewer 1 Report

The authors performed all suggested explanations. 

Author Response

Dear Reviewer,

Thank you for the last report. According to your suggestions we did our best in correcting spelling/style of the manuscript.

Regards,

Radosław Czernych
